# The Portal Venous Pulsatility Index and Main Portal Vein Diameter as Surrogate Markers for Liver Fibrosis in Nonalcoholic Fatty Liver Disease and Metabolic-Dysfunction-Associated Steatotic Liver Disease

**DOI:** 10.3390/diagnostics14040393

**Published:** 2024-02-11

**Authors:** Jaejun Lee, Seungmyeon Choi, Seong-Hyun Cho, Hyun Yang, Pil-Soo Sung, Si-Hyun Bae

**Affiliations:** 1The Catholic University Liver Research Center, College of Medicine, The Catholic University of Korea, Seoul 06591, Republic of Korea; pwln0516@gmail.com (J.L.); oneggu@naver.com (H.Y.); pssung@catholic.ac.kr (P.-S.S.); 2Department of Internal Medicine, Armed Forces Goyang Hospital, Goyang 10267, Republic of Korea; joyjoy1@naver.com; 3Department of Radiology, Armed Forces Goyang Hospital, Goyang 10267, Republic of Korea; choi12111@gmail.com; 4Division of Gastroenterology and Hepatology, Department of Internal Medicine, College of Medicine, Eunpyeong St. Mary’s Hospital, The Catholic University of Korea, Seoul 03383, Republic of Korea; 5Division of Gastroenterology and Hepatology, Department of Internal Medicine, College of Medicine, Seoul St. Mary’s Hospital, The Catholic University of Korea, Seoul 06591, Republic of Korea

**Keywords:** portal venous pulsatility index, liver fibrosis, nonalcoholic fatty liver disease, metabolic-dysfunction-associated steatotic liver disease, biomarker

## Abstract

(1) Background: Despite numerous noninvasive methods for assessing liver fibrosis, effective ultrasound parameters remain limited. We aimed to identify easily measurable ultrasound parameters capable of predicting liver fibrosis in patients with nonalcoholic fatty liver disease (NAFLD) and metabolic-dysfunction-associated steatotic liver disease (MASLD); (2) Methods: The data of 994 patients diagnosed with NAFLD via ultrasound at the Armed Forces Goyang Hospital were retrospectively collected from June 2022 to July 2023. A liver stiffness measurement (LSM) ≥ 8.2 kPa was classified as significant fibrosis. Liver steatosis with cardiometabolic risk factors was defined as MASLD. Two ultrasound variables, the portal venous pulsatility index (VPI) and main portal vein diameter (MPVD), were measured; (3) Results: Of 994 patients, 68 had significant fibrosis. Significant differences in VPI (0.27 vs. 0.34, *p* < 0.001) and MPVD (10.16 mm vs. 8.98 mm, *p* < 0.001) were observed between the fibrotic and non-fibrotic groups. A logistic analysis adjusted for age and body mass index (BMI) revealed that only VPI (OR of 0.955, *p* = 0.022, VPI on a 0.01 scale) and MPVD (OR of 1.501, *p* < 0.001) were significantly associated with significant liver fibrosis. In the MASLD cohort (*n* = 939), VPI and MPVD were associated with significant fibrosis. To achieve better accuracy in predicting liver fibrosis, we established a nomogram that incorporated MPVD and VPI. The established nomogram was validated in the test cohort, yielding an area under the receiver operating characteristic curve of 0.821 for detecting significant liver fibrosis; (4) Conclusions: VPI and MPVD, as possible surrogate markers, are useful in predicting significant fibrosis in patients with NAFLD and MASLD.

## 1. Introduction

Nonalcoholic fatty liver disease (NAFLD) is a global health concern, progressively contributing to the prevalence of liver cirrhosis and liver-related mortalities [1,2]. The prognosis of NAFLD varies, primarily dependent on disease severity, specifically the stage and presence of nonalcoholic steatohepatitis (NASH) [3,4,5]. Notably, NAFLD with a fibrosis stage of F2 or higher is a prominent determinant of unfavorable liver-related outcomes [6]. Therefore, identifying patients with NAFLD and fibrosis stage F2 or higher is imperative for implementing individualized and effective management and surveillance strategies.

Traditionally, a liver biopsy has been the gold standard for evaluating fibrosis. However, its invasiveness and associated risks render it unappealing to patients with NAFLD. As an alternative, various non-invasive methods, including serum- and imaging-based biomarkers, have been developed to assess significant or advanced fibrosis [7,8]. Nevertheless, most serum biomarkers have limited diagnostic performance in predicting liver fibrosis, particularly in younger populations [9,10]. Additionally, imaging-based non-invasive biomarkers are often costly and may not be readily available in developing countries or within the primary care setting of well-developed nations [11]. Consequently, there remains a demand for alternative tools to assess the fibrosis stage.

The portal venous pulsatility index (VPI) is used to quantify the pulsatility of blood flow in the portal vein and is a point-of-care measurement. It is calculated using the following formula using the highest and lowest velocity of portal vein flow: Vmax-Vmin/Vmax [12]. Studies have been conducted to determine the utility and significance of VPI in liver diseases and several have shown that VPI may be lower in chronic liver disease compared to healthy subjects [13,14,15,16]. Furthermore, a subset of studies has demonstrated a reduction in VPI as the degree of steatosis increases, particularly in NAFLD [14,15,17]. Baikpour et al. also conducted a study on NAFLD cohorts confirmed by biopsy. They found that VPI was lower in patients with fibrosis stage F2 or higher compared to those with F0 or F1 [18]. This finding was corroborated by a subsequent study conducted by Hamed et al., which also observed a lower VPI among patients with NAFLD and fibrosis stage F2 or greater [19]. Lu et al. presented contrasting results, showing no significant differences in VPI across different fibrosis stages [20]. Therefore, the potential of VPI as a predictive marker for liver fibrosis remains controversial.

As previously highlighted, the discrimination of patients with NAFLD and fibrosis stage F2 or higher is a pivotal aspect of NAFLD management. Therefore, there remains an unmet need for simpler point-of-care methods to identify these patients. Given the ongoing controversy surrounding the correlation between VPI and liver fibrosis in NAFLD, we investigated the potential role of VPI as a predictive marker of liver fibrosis, particularly in stage F2 or higher. Furthermore, considering the recent introduction of the term “metabolic dysfunction-associated steatotic liver disease” (MASLD) through expert consensus via the Delphi process, our study sought to evaluate VPI as a predictive marker for MASLD [21].

## 2. Materials and Methods

### 2.1. Patients

We conducted a retrospective study using data from patients who visited Goyang Armed Forces Hospital between June 2022 and July 2023. The eligible participants underwent both abdominal ultrasonography and vibration-controlled transient elastography (VCTE). The inclusion criteria were the following: (1) patients diagnosed with NAFLD using abdominal ultrasonography and medical history, (2) age > 18, and (3) patients who underwent VCTE within one month before or after the abdominal ultrasound. Patients were excluded from this study for the following reasons: (1) alanine aminotransferase (ALT) levels exceeding five times the upper limit of normal due to the potential for an overestimation of liver stiffness measurement (LSM) [22], (2) excessive alcohol consumption (defined as >210 g per week for men and 140 g per week for women), (3) history of heart failure or coronary artery diseases, and (4) liver cirrhosis. This study was approved by the Institutional Review Board of the Korean Armed Forces Medical Command (AFMC 2023-08-006-002) and was conducted in accordance with the Declaration of Helsinki. The informed consent requirement was waived owing to the study’s retrospective nature.

### 2.2. Transient Elastography

Controlled attenuation parameter (CAP) and liver stiffness measurements with VCTE were conducted using FibroScan^®^502 (Echosens, Paris, France). All patients were instructed to fast for at least 8 h before LSM measurement. The result was considered reliable if it consisted of ten valid measurements with a success rate exceeding 60%. For those with LSM measurements >7.1 kPa, a reliable measurement was defined as an interquartile range (IQR) to median ratio <30% and was included for the analysis [23].

### 2.3. Sonographic Variables

Steatosis was assessed using abdominal ultrasonography (S1000, Siemens, Munich, Germany). The patients maintained at least 8 h of fasting when ultrasonography was performed. In our institution, VPI and the main portal vein diameter (MPVD) are routinely measured during ultrasonography sessions. To calculate VPI, the right portal vein was located in the intercostal view. The spectral Doppler mode was selected to measure the highest and lowest velocities of portal vein flow (Figure 1). An experienced sonographer, blinded to other clinical variables, such as LSM and laboratory findings, performed the VPI measurements. The MPVD was measured in the subcostal view, where the inferior vena cava crossed the MPVD in the longitudinal sonographic plane. The diameter was measured from the inner wall to the inner wall while the patient maintained a quiet respiratory state. In cases where ultrasound parameters were measured multiple times, the median value was adopted for the analysis. To assess interobserver variability, two additional experienced sonographers (S.M.C. and S.H.C.) independently reviewed the sonographic images and measured the VPI and MPVD without knowledge of the measurements made by the other sonographers.

### 2.4. Definitions

MASLD was defined within the NAFLD patient population if one or more of the following cardiometabolic risk factors (CMRFs) were present [21]: (1) body mass index (BMI) ≥ 23 kg/m^2^ or waist circumference ≥ 90 cm for men and ≥85 cm for women [24]; (2) fasting serum glucose ≥ 100 mg/dL or HbA1c ≥ 5.7% or type 2 diabetes or treatment for type 2 diabetes; (3) blood pressure ≥ 130/85 mmHg or antihypertensive drug treatment; (4) triglycerides ≥ 150 mg/dL or lipid-lowering drug use; (5) high-density lipoprotein cholesterol ≤ 40 mg/dL for men and ≤50 mg/dL for women or lipid-lowering drug use. Significant fibrosis (F2 or greater in METAVIR) was defined as an LSM ≥ 8.2 kPa according to the cut-off value suggested by Eddowes et al. [25]. Patients with LSM ≥ 8.2 kPa were classified into the SF (+) group, while those falling below this threshold were categorized into the SF (−) group. Other determinant values for significant fibrosis were also adopted for the sensitivity analysis, namely LSM > 7.0 kPa and a Fibroscan-AST (FAST) score > 0.67 [26,27]. The degree of steatosis was stratified using the CAP value as follows: S1 = CAP score < 310 dB/m, S2 = CAP score between 310 and 331 dB/m, and S3 = CAP score > 331 dB/m [25]. The diagnosis of other metabolic conditions, such as diabetes mellitus (DM), hypertension (HTN), and dyslipidemia, was based on medical history or laboratory findings. Dyslipidemia was defined in accordance with the criteria established by the Korean Society of Dyslipidemia and Atherosclerosis [28]. Non-invasive biomarkers, including the Fibrosis-4 (FIB-4) score, NAFLD Fibrosis Score (NFS), AST to platelet ratio index (APRI) score, and hepatic steatosis index (HSI), were calculated and subsequently factored into the comparative analysis [29,30,31,32].

### 2.5. Allocation of Derivation and Validation Cohorts

To construct the nomogram, we randomly allocated the entire cohort into test and training cohorts (2:1). This allocation was performed using R statistical software (version 4.0.3; R Foundation Inc., Vienna, Austria; http://cran.r-project.org, accessed on 3 September 2023), irrespective of baseline characteristics such as age, sex, BMI, or other laboratory variables.

### 2.6. Statistical Analysis

All statistical analyses were performed using R statistical software (version 4.0.3; R Foundation Inc., Vienna, Austria; http://cran.r-project.org, accessed on 3 September 2023). For continuous variables, either the Student’s *t*-test or Wilcoxon rank-sum test was performed, and results were presented as mean values with standard deviations for normally distributed variables, whereas the median with interquartile range (IQR) was used for non-normally distributed variables. The normality of each variable was assessed using the Shapiro–Wilk test. Categorical variables were analyzed using either the chi-square test or Fisher’s exact test depending on the sample size. A linear regression analysis was performed to identify the relationship between continuous variables, and a logistic regression analysis was employed to identify factors associated with significant fibrosis, with results presented as odds ratios (ORs), 95% confidence intervals (95% CIs), and corresponding p values. A receiver operating characteristic curve (ROC) was used to visually depict and demonstrate the diagnostic performance of each variable or score. Interclass correlation coefficients were calculated to assess interobserver variations, with coefficients exceeding 0.9 indicating excellent reliability and coefficients falling between 0.75 and 0.9 denoting good reliability. Significance was set at *p* < 0.05.

## 3. Results

### 3.1. Baseline Characteristics

A total of 1142 patients with a fatty liver were considered for inclusion in our study. Of those, 994 patients with NAFLD were included in the final analysis (Figure 2). The baseline patient characteristics are summarized in Table 1. The median age was 21.0 and the majority were male (98.0%). The mean BMI of the study population was 27.4 kg/m^2^ and the prevalence of metabolic comorbidities, such as obesity (67.9%), DM (4.4%), HTN (17.7%), and dyslipidemia (35.5%), was also assessed. Regarding the number of CMRFs present in each patient, 37.3% had one CMRF, 29.5% had two CMRFs, and 27.7% had three or more CMRFs.

Values are presented as mean ± standard deviation, median (interquartile range), or number (%). NAFLD was stratified by using liver stiffness measurement with the threshold of 8.2 kPa. Abbreviations: ALT, alanine aminotransferase; AST, aspartate aminotransferase; BMI, body mass index; CAP, controlled attenuation parameter; CMRF, cardiometabolic risk factor; DM, diabetes mellitus; GGT, gamma-glutamyl transferase; HTN, hypertension; INR, international normalized ratio; IQR, interquartile range; LSM, liver stiffness measurement; Med, median; NAFLD, nonalcoholic fatty liver disease; PLT, platelet count; PT, prothrombin time; TB, total bilirubin.

Patients were stratified into SF (+) (*n* = 68) and SF (−) (*n* = 926) groups by LSM of 8.2 kPa. A comparison of these two groups revealed a significantly higher BMI in the SF (+) group compared to the SF (−) group (33.0 kg/m^2^ vs. 27.0 kg/m^2^, *p* < 0.001). No significant differences were observed in terms of age, male sex, or prevalence of HTN and DM between the two groups. The SF (+) group had a higher proportion of patients with dyslipidemia, although this difference was not statistically significant (47.1% vs. 34.7%, *p* = 0.054). Regarding laboratory findings, ALT, aspartate aminotransferase (AST), and gamma-glutamyl transferase (GGT) were all elevated in the SF (+) group compared to the SF (−) group. The median CAP score and median LSM for the entire cohort were 278.0 dB/m and 4.8 kPa, respectively, with both values being significantly higher in the SF (+) group compared to the SF (−) group (median CAP of 340.0 vs. 271.0 dB/m, *p* < 0.001; median LSM of 9.1 vs. 4.7, *p* < 0.001; SF (+) and SF (−) group, respectively).

### 3.2. Comparison of Non-Invasive Markers in SF (+) and SF (−) NAFLD Groups

Non-invasive markers known to predict liver fibrosis or steatosis were compared between the SF (+) and SF (−) groups (Table 2). The HSI, designed to predict liver steatosis, was higher in the SF (+) group compared to the SF (−) group (49.12 ± 6.33 vs. 39.79 ± 7.70, *p* < 0.001). Fibrosis markers were also assessed. Both the NFS (−3.43 ± 1.07 vs. −3.82 ± 1.13, *p* = 0.006) and the APRI score (0.55 ± 0.26 vs. 0.32 ± 0.23, *p* < 0.001) showed significant differences, with higher values observed in the SF (+) group. In contrast, the FIB-4 score displayed no significant difference between the two groups (0.49 ± 0.19 vs. 0.45 ± 0.26, *p* = 0.173).

Additionally, sonographic variables, namely MPVD and VPI, were compared according to the fibrosis stage and degree of steatosis. In terms of MPVD, the SF (+) group exhibited MPVD of 10.16 ± 1.62 mm, significantly higher than the SF (−) group, which had an MPVD of 8.98 ± 1.30 mm (*p* < 0.001). In contrast, VPI was 0.27 ± 0.12 in the SF (+) group and 0.34 ± 0.11 in the SF (−) group, indicating a significantly lower VPI in the SF (+) group (*p* < 0.001) (Figure 3A).

### 3.3. VPI According to the CAP Score and LSM

In addition to the differences in VPI between the NAFLD risk groups stratified by LSM, VPI was also compared between patients with NAFLD and different degrees of steatosis (Figure 3B). Consequently, VPI was 0.36 ± 0.11 in S1, 0.29 ± 0.07 in S2, and 0.28 ± 0.07 in S3, demonstrating a decreasing trend as the steatosis worsened. Significant differences were observed between S1 and S2 (*p* < 0.001) and between S1 and S3 (*p* < 0.001), whereas no statistical difference was identified between S2 and S3 (*p* = 0.189). Furthermore, a linear regression analysis was used to assess the correlation between VPI and the CAP score, as well as LSM (Appendix A). Regarding the relationship between VPI and LSM, an inverse correlation was detected, indicating that LSM decreases as VPI increases (r = −0.163, *p* < 0.001). A similar inverse correlation was found between the CAP score and VPI, with an r value of −0.439 (*p* < 0.001).

### 3.4. Factors Associated with Significant Fibrosis in NAFLD

A logistic regression analysis was used to identify factors associated with significant fibrosis. Factors, including sonographic variables, age, and metabolic diseases, were evaluated for their association with significant fibrosis (Table 3). In the univariate analysis, VPI, MPVD, and dyslipidemia were found to be factors associated with significant fibrosis. In the age-adjusted model, MPVD (odds ratio [OR] of 1.779, 95% confidence interval [CI] of 1.498–2.111, *p* < 0.001), BMI (OR of 1.389, 95% CI of 1.296–1.488, *p* < 0.001), DM (OR of 2.614, 95% CI of 1.013–6.742, *p* = 0.047), and dyslipidemia (OR of 1.819, 95% CI of 1.089–3.039, *p* = 0.022) were all positively correlated with significant fibrosis. In contrast, VPI was negatively correlated with significant fibrosis in the age-adjusted model (OR of 0.910, 95% CI of 0.878–0.944, *p* < 0.001, VPI on a 0.01 scale). Subsequently, a multivariate analysis was conducted using age and BMI ≥ 30 kg/m^2^ as covariates. MPVD showed a significant correlation with fibrosis (OR of 1.501, 95% CI of 1.243–1.813, *p* < 0.001). Consistent with the results of other models, VPI showed an inverse correlation with significant fibrosis (OR of 0.955, 95% CI of 0.919–0.993, *p* = 0.022; VPI, 0.01).

### 3.5. Subgroup Analysis in Patients with Age over 30

Given that most of the study cohorts were <30, a subgroup analysis of patients aged >30 was also conducted. A total of 184 patients were included in this category; 13 were classified as having significant fibrosis. The baseline characteristics of this subgroup are presented in Appendix A. In summary, the median age was 40.0, and 178 patients (96.7%) were male. BMI, AST, and ALT levels were higher in the SF (+) group. Subsequently, a univariate logistic regression analysis was performed on this subgroup to identify the factors associated with significant fibrosis (Appendix A). Overall, MPVD and BMI were risk factors for significant fibrosis. Once again, VPI exhibited an inverse correlation with significant fibrosis (OR of 0.822, 95% CI of 0.721–0.938, *p* < 0.001, VPI on a 0.01 scale), which is consistent with the results observed in the entire cohort.

### 3.6. Sensitivity Analysis Using Different Determinant Values for Significant Fibrosis

A sensitivity analysis was conducted using two distinct values to determine the significance of fibrosis (Appendix A). First, factors associated with significant fibrosis were assessed using a FAST score >0.67 as the criterion for significant fibrosis. In the age-adjusted model, with age as a covariate, VPI (OR of 0.885, 95% CI of 0.846–0.925, *p* < 0.001, VPI on a 0.01 scale), MPVD (OR of 1.811, 95% CI of 1.506–2.177, *p* < 0.001), BMI ≥ 30 kg/m^2^ (OR of 8.216, 95% CI of 4.449–15.172, *p* < 0.001), and dyslipidemia (OR of 3.657, 95% CI of 2.069–6.463, *p* < 0.001) were associated with significant fibrosis. When BMI ≥ 30 kg/m^2^ was added as a covariate (Model 2), associations with significant fibrosis persisted for VPI (OR of 0.916, 95% CI of 0.874–0.960, *p* = 0.002, VPI on a 0.01 scale), MPVD (OR of 1.578, 95% CI of 1.296–1.920, *p* < 0.001), and dyslipidemia (OR of 2.461, 95% CI of 1.355–4.469, *p* = 0.003).

Subsequently, an LSM threshold of >7.0 kPa was employed to define significant fibrosis, and the variables were evaluated for their associations. In the age-adjusted model, VPI, MPVD, BMI ≥ 30 kg/m^2^, and dyslipidemia were all significantly associated with fibrosis. However, when BMI ≥ 30 kg/m^2^ was added as a covariate, only VPI (OR of 0.949, 95% CI of 0.922–0.977, *p* < 0.001, VPI on a 0.01 scale) and MPVD (OR of 1.706, 95% CI of 1.451–2.007, *p* < 0.001) were associated with significant fibrosis.

### 3.7. Sensitivity Analysis in Patients with MASLD

Among the cohort of 994 patients diagnosed with NAFLD, 939 presented with cardiometabolic risk factors categorized under the MASLD designation. In the MASLD subgroup, 68 patients showed significant fibrosis. The examination of variables for their association with significant fibrosis in the MASLD group employed two previously introduced multivariate models, as outlined in Appendix A. In the age-adjusted model, VPI, MPVD, and BMI were identified as factors associated with significant fibrosis. Subsequently, in the multivariate model, only VPI (OR of 0.951, 95% CI of 0.924–0.979, *p* = 0.001, VPI on a 0.01 scale) and MPVD (OR of 1.687, 95% CI of 1.435–1.984, *p* < 0.001) emerged as the two factors found to be significantly associated with significant fibrosis in the MASLD patient subset.

### 3.8. Developing Nomogram for Predicting Liver Fibrosis

To develop a predictive model using ultrasonographic variables for significant fibrosis, the entire NAFLD cohort was randomly divided into training and test cohorts in a 2:1 ratio. No significant differences in body measurements, prevalence of metabolic diseases, or laboratory findings were detected (Appendix A). A nomogram was established from the training cohort using two variables: VPI and MPVD. The MPVD score ranged from 0 to 100 as the MPVD increased from 5 to 15 mm, whereas the VPI score ranged from 0 to 97 as the VPI decreased from 0.9 to 0.0 (Appendix A). Overall, the total score using the nomogram ranged from a minimum score of 34.1 to a maximal score of 169.9 in the test cohort. Next, the predictive performance of the established nomogram and sonographic variables for significant liver fibrosis was evaluated. Figure 4 shows the receiver operating characteristic curves for VPI, MPVD, and the nomogram for predicting significant liver fibrosis. Consequently, the area under the receiver operating characteristic curve (AUROC) for MPVD was 0.731 (95% CI: 0.616–0.846), and for VPI, it was 0.770 (95% CI: 0.681–0.859). Regarding the established nomogram, a larger AUROC of 0.821 (95% CI: 0.735–0.908) was noted compared to the previous two variables. Furthermore, the nomogram was compared to existing non-invasive biomarkers, specifically the FIB-4 score and NFS. Consequently, the nomogram demonstrated superior predictive performance compared to both FIB-4 (*p* = 0.002) and NFS (*p* = 0.002).

Using the provided ultrasound parameters, an algorithm for assessing liver fibrosis can be proposed, with respective thresholds of 120 for the nomogram (sensitivity of 71.9%, specificity of 74.7%) and 0.26 for the VPI (sensitivity of 59.5%, specificity of 77.8%) (Figure 5).

### 3.9. Interobserver and Intraobserver Variations Regarding Ultrasonographic Variables

To address and mitigate the potential bias caused by measurements taken by a single observer, the interobserver variation was calculated for VPI and MPVD. Two other experienced sonographers independently reviewed the sonographic images using a Picture Archiving and communication system (PACS). Regarding VPI, the interobserver variability, as measured using intraclass correlation coefficients, was calculated to be 0.932 (95% CI: 0.900–0.952), indicating excellent reliability. The interobserver variability for MPVD was 0.798 (95% CI: 0.376–0.911), demonstrating good reliability. In addition, intraobserver variability was assessed. A total of 107 and 166 sonographic records, each measured more than once for VPI and MPVD, respectively, were identified. Utilizing these records, intraobserver variability was determined to be 0.959 (95% CI: 0.920–0.976) for VPI and 0.820 (95% CI: 0.450–0.920) for MPVD.

## 4. Discussion

In this study, we evaluated the sonographic parameter VPI as a potential marker for assessing NAFLD status. VPI, which can be conveniently measured using spectral Doppler mode ultrasound, exhibited notable differences between the SF (−) and SF (+) groups, with lower values observed in patients with significant liver fibrosis. Furthermore, the VPI showed a decreasing trend with worsening steatosis. Most importantly, we identified VPI as a novel factor associated with significant liver fibrosis and demonstrated an inverse correlation. Our study revealed an association between MPVD and fibrosis. These results were consistently reproduced in patients diagnosed with MASLD, highlighting VPI and MPVD as significant factors for fibrosis in individuals with MASLD. Collectively, these findings suggest that both VPI and MPVD have the potential to distinguish patients with significant fibrosis in NAFLD and MASLD, extending their relevance beyond NAFLD alone.

The Doppler flow pattern in chronic liver disease exhibits various forms corresponding to the severity of disease. In addition, the Doppler flow of the portal vein is influenced by various factors, such as right heart failure or arteriovenous shunting [33,34]. Liver cirrhosis, in particular, might show diverse patterns of portal vein flow such as retrograde flow, pulsatile flow, or even antegrade flow with low portal venous velocity, contingent on the presence of portal hypertension or arteriovenous shunts [35]. Conversely, the pattern of VPI in a non-cirrhotic liver is typically monophasic and characterized by antegrade flow [33]. Hence, it is essential to distinguish between NAFLD with F0 to F3 and cirrhosis in terms of a VPI analysis due to distinct pathophysiological mechanisms underlying portal venous flow, which is why cirrhotic livers were deliberately excluded from our study.

As stated earlier, we found that VPI was lower in patients with significant fibrosis. Several potential mechanisms can explain this observation. First, fatty infiltration in the liver can restrict the space available for hepatic vasculature, leading to reduced portal venous velocity and a decrease in VPI [36]. BMI could impact vascular compliance by increasing abdominal pressure, thereby resulting in a lower VPI value [12]. Additionally, fibrosis in the liver parenchyma could cause sinusoidal inflammation and injury, which in turn increases the resistance of the portal vein and lowers the portal vein velocity. Similarly, Barakat et al. proposed a mechanism suggesting that pathological liver fibrosis reduces the transmission of atrial pressure to the liver, subsequently reducing VPI [37].

Prior to our study, several studies have explored the relationship between VPI and chronic liver diseases, and our findings align with these investigations [13,14,15,16,17]. For instance, Baikpour et al. studied 123 subjects with biopsy-proven NAFLD and reported a mean VPI of 0.28. Notably, they found that patients with NAFLD and significant fibrosis (F2 or greater) exhibited a lower VPI value of 0.19 compared to patients with F0 or F1 who had a VPI of 0.32 [18]. Our study corroborates these findings as we observed a VPI of 0.34 in the SF (−) group, which is consistent with previous reports. However, a slight variation emerged in patients with NAFLD and significant fibrosis, as our results indicated a VPI of 0.27, which was slightly higher than the value reported by Baikpour et al. Nevertheless, these variations were attributed to the relatively milder degree of liver fibrosis in the SF (+) group in our study, as the majority of patients had LSM of <12 kPa. In the same context, other studies have reported similar results to ours in relation to VPI of the SF (+) group, ranging from 0.20 to 0.26 [16,17,19,20]. These consistent measurements across studies underscore the reliability of VPI as an ultrasonographic parameter.

In this study, we created a nomogram to enhance the prediction of significant liver fibrosis by incorporating both VPI and MPVD as variables. MPVD and its association with advanced fibrosis has been studied in multiple investigations, consistently revealing higher MPVD values in patients with advanced fibrosis [38]. Gathering these two variables, the nomogram showed enhanced predictability with AUROC of 0.821, surpassing the performance of existing non-invasive markers such as the FIB-4 score or NFS. However, the established nomogram requires additional external validation before it can be considered a viable substitute biomarker.

Despite the numerous studies on VPI and NAFLD conducted over the past two decades, our study is unique in various aspects. First, the present study included the largest number of patients with NAFLD compared to previous studies, with the largest prior study including fewer than 200 patients with NAFLD [19]. Moreover, by using LSM as an indicator for liver fibrosis, a linear relationship between liver stiffness and VPI could be described in this study. Additionally, our study excluded liver cirrhosis from the analysis because the portal venous flow pattern can vary in cirrhosis due to severe portal hypertension or arteriovenous shunting. Cirrhosis can be determined using certain findings on routine abdominal ultrasonography, such as the surface nodularity, heterogeneous echotexture of the liver parenchyma, and changes in liver volume distribution [39,40]. Therefore, excluding cirrhosis from VPI measurements reduces the risk of bias in the result, while maintaining its purpose of addressing the unmet need for differentiating significant or advanced fibrosis at the point of care. Moreover, we conducted a comprehensive analysis of MASLD, a term recently suggested as a substitute for NAFLD. To the best of our knowledge, this study is the first to demonstrate the potential of VPI as a marker of liver fibrosis in MASLD. Lastly, we integrated MVPD with VPI in predicting significant fibrosis, showcasing the potential of this aggregated biomarker as a superior surrogate marker compared to each individually.

Although our study had several strengths, it also had several limitations. Histopathological data were absent in this study, which might introduce the potential misdiagnosis of significant fibrosis. Although LSM measured using transient elastography has proven to be highly reliable in identifying significant fibrosis, and sensitivity analyses using different cut-off values and tools have been conducted to mitigate bias, data from biopsy-proven studies could offer a more definitive understanding of the correlation between VPI and liver fibrosis without controversy [25]. A comparison of predictive performance between LSM and VPI in biopsy-proven cohorts could enhance the significance of VPI in predicting significant fibrosis. In addition, the study population was skewed toward a younger demographic, with the majority <30, and included a limited number of female patients. Although a sensitivity analysis using patients >30 yielded results consistent with those of the entire cohort, the generalizability of our findings to broader populations needs to be further validated. Furthermore, despite the promising interobserver variabilities of ultrasound parameters in our study, measurements by other sonographers were not conducted in separate sessions and were solely based on static photos, introducing certain limitations regarding the calculated interobserver variabilities. Finally, the single-center design could raise concerns regarding the reliability of the results, particularly those of the nomogram.

In conclusion, our study demonstrated that VPI could predict significant fibrosis in patients with NAFLD and MASLD. Given that both VPI and MPVD can be easily measured during routine abdominal ultrasound sessions, our findings have the potential to benefit clinicians in assessing the status of chronic liver disease, particularly in primary care settings where VCTE is not commonly available. However, considering the limitations of the present study, further studies are warranted to strengthen these correlations and reach a consensus on the diagnostic utility of these sonographic parameters.

## Figures and Tables

**Figure 1 diagnostics-14-00393-f001:**
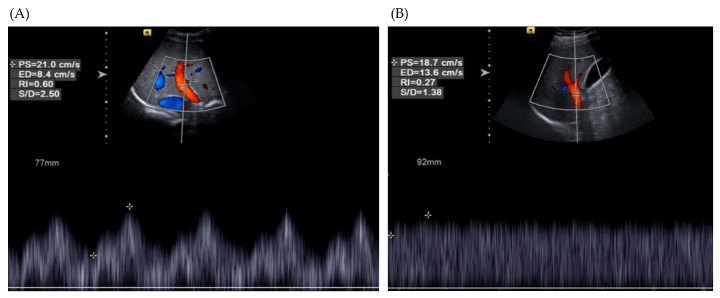
Measurement of portal venous pulsatility index. Right portal vein was identified in the intercostal view. (**A**) A 21-year-old nonalcoholic fatty liver disease patient with a liver stiffness measurement (LSM) of 5.0 kPa, showing VPI of 0.6. (**B**) A 30-year-old man with an LSM of 8.4 kPa, showing VPI of 0.27. Abbreviations: LSM, liver stiffness measurement; VPI, portal venous pulsatility index.

**Figure 2 diagnostics-14-00393-f002:**
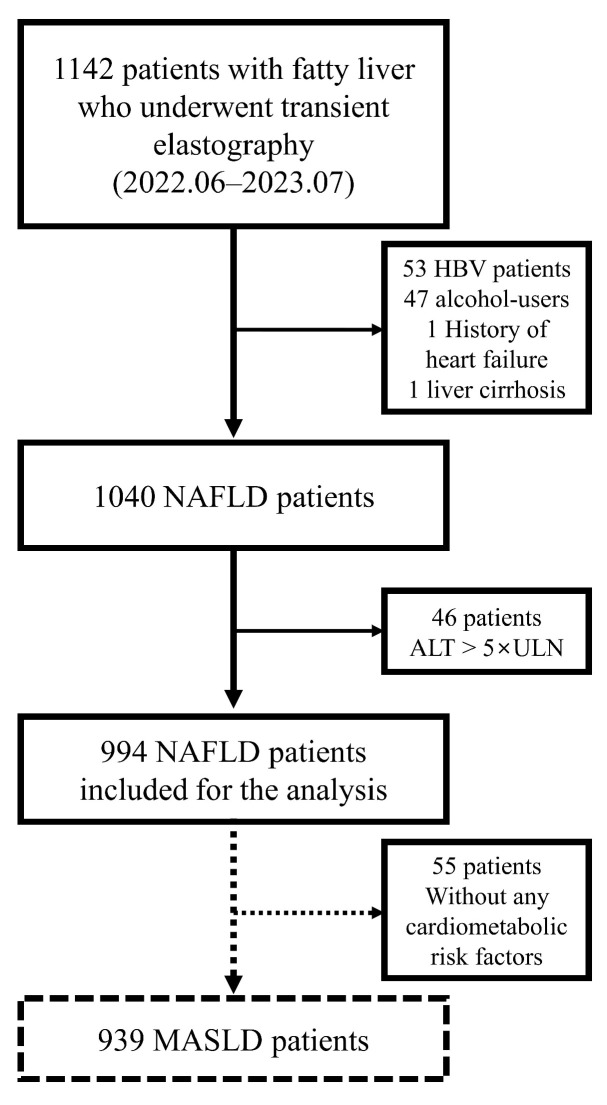
Patient selection flowchart. Abbreviations: ALT, alanine aminotransferase; NAFLD, nonalcoholic fatty liver disease; MASLD, metabolic-dysfunction-associated steatotic liver disease; ULN, upper limit of normal.

**Figure 3 diagnostics-14-00393-f003:**
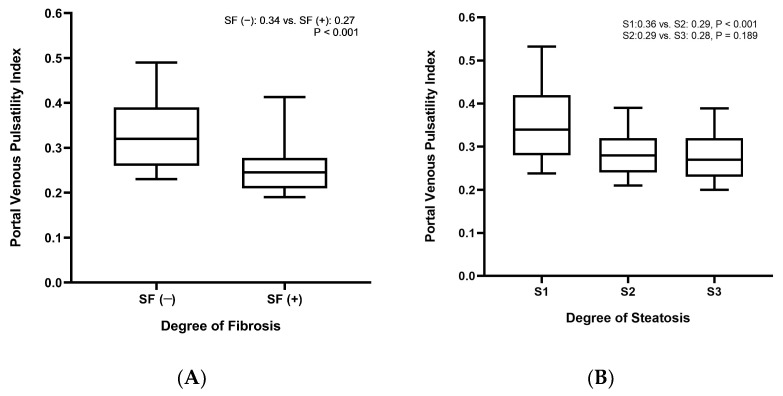
Boxplot of portal venous pulsatility index stratified by (**A**) fibrosis stage and (**B**) steatosis degree.

**Figure 4 diagnostics-14-00393-f004:**
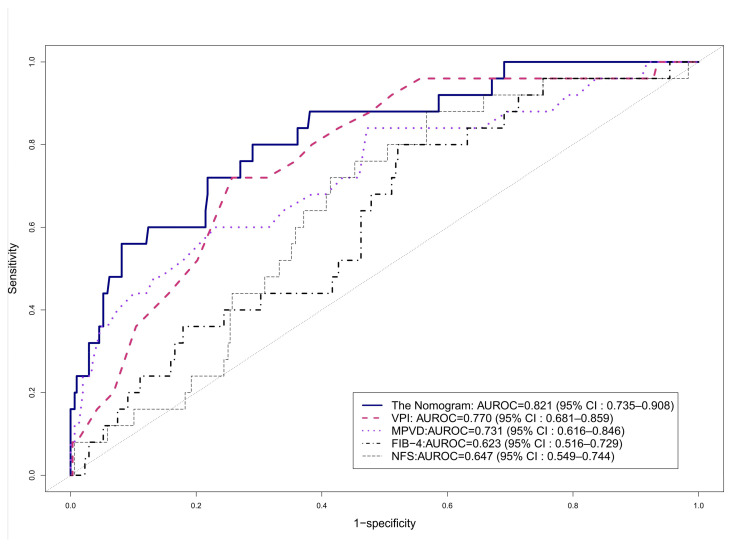
ROC curves for significant fibrosis prediction model. Abbreviations: AUROC, area under the receiver operating characteristics; FIB-4, Fibrosis-4; MPVD, main portal vein diameter; NAFLD, nonalcoholic fatty liver disease; NFS, NAFLD Fibrosis Score; ROC, receiver operating characteristics; VPI, portal venous pulsatility index.

**Figure 5 diagnostics-14-00393-f005:**
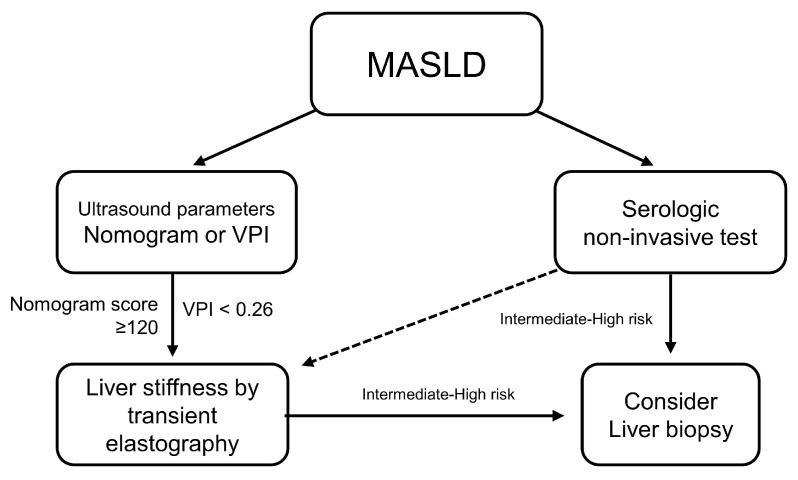
Proposed use of ultrasound parameters. Abbreviations: MASLD, metabolic-dysfunction-associated liver disease; VPI, portal venous pulsatility index.

**Table 1 diagnostics-14-00393-t001:** Baseline characteristics.

	Total (*n* = 994)	SF (−) Group (*n* = 926)	SF (+) Group (*n* = 68)	*p* Value
Male sex	974 (98.0)	907 (97.9)	67 (98.5)	1.000
Age	21.0 (20.0, 26.0)	21.0 (20.0, 26.0)	21.0 (20.0, 25.5)	0.333
Height (cm)	174.6 ± 5.9	174.6 ± 5.8	175.8 ± 6.5	0.107
Weight (kg)	83.8 ± 15.0	82.4 ± 14.1	102.1 ± 14.9	<0.001
BMI (kg/m^2^)	27.4 ± 4.4	27.0 ± 4.1	33.0 ± 4.1	<0.001
Obesity (BMI ≥ 25 kg/m^2^)	675 (67.9)	611 (66.0)	64 (94.1)	<0.001
DM	44 (4.4)	38 (4.1)	6 (8.8)	0.128
HTN	176 (17.7)	164 (17.7)	12 (17.6)	1.000
Dyslipidemia	353 (35.5)	321 (34.7)	32 (47.1)	0.054
Number of CMRFs				<0.001
1	371 (37.3)	358 (38.7)	13 (19.1)	
2	293 (29.5)	269 (29.0)	24 (35.3)	
≥3	275 (27.7)	244 (26.3)	31 (45.6)	
PLT (1000/μL)	266.0 (233.0, 301.0)	265.0 (233.0, 300.0)	276.0 (236.5, 302.8)	0.378
AST (IU/L)	28.7 (20.5, 40.6)	27.9 (20.4, 38.6)	59.3 (41.5, 71.1)	<0.001
ALT (IU/L)	47.0 (24.0, 83.0)	44.5 (23.0, 77.0)	132.5 (73.3, 165.2)	<0.001
TB (mg/dL)	0.8 (0.6, 1.0)	0.8 (0.6, 1.0)	0.8 (0.6, 0.9)	0.132
GGT (U/L)	40.0 (23.0, 70.0)	38.0 (22.0, 69.0)	62.0 (44.0, 95.0)	<0.001
Albumin (mg/dL)	4.9 (4.7, 5.1)	4.9 (4.7, 5.1)	4.9 (4.8, 5.1)	0.942
PT (INR)	1.01 (0.97, 1.05)	1.01 (0.97, 1.05)	0.99 (0.96, 1.02)	0.005
CAP (dB/m)	278.0 (244.0, 329.0)	271.0 (242.0, 324.0)	340.0 (313.0, 363.0)	<0.001
LSM (kPa)	4.8 (4.0, 5.9)	4.7 (4.0, 5.6)	9.1 (8.7, 10.6)	<0.001
Med/IQR for LSM	14.0 (9.0, 18.0)	13.0 (9.0, 18.0)	16.0 (11.0, 20.0)	0.011

**Table 2 diagnostics-14-00393-t002:** Comparison of non-invasive markers in two groups.

	SF (−) Group (*n* = 926)	SF (+) Group (*n* = 68)	*p* Value
VPI	0.34 ± 0.11	0.27 ± 0.12	<0.001
MPVD (mm)	8.98 ± 1.30	10.16 ± 1.62	<0.001
FIB-4	0.45 ± 0.26	0.49 ± 0.19	0.173
NFS	−3.82 ± 1.13	−3.43 ± 1.07	0.006
APRI	0.32 ± 0.23	0.55 ± 0.26	<0.001
HSI	39.79 ± 7.70	49.12 ± 6.33	<0.001

Values are presented as mean ± standard deviation. Abbreviations: APRI, AST to platelet ratio index; FIB-4, Fibrosis-4; HSI, hepatic steatosis index; MPVD, main portal vein diameter; NFS, NAFLD Fibrosis Score; VPI, portal venous pulsatility index.

**Table 3 diagnostics-14-00393-t003:** Univariate and multivariate analysis of factors associated with significant fibrosis in NAFLD.

	Crude OR (95% CI)	*p*	Age-Adjusted OR (95% CI)	*p*	Multivariable-Adjusted OR (95% CI)	*p*
VPI (0.01 scale)	0.920 (0.889–0.952)	<0.001	0.910 (0.878–0.944)	<0.001	0.955 (0.919–0.993)	0.022
MPVD	1.779 (1.498–2.112)	<0.001	1.779 (1.498–2.111)	<0.001	1.501 (1.243–1.813)	<0.001
BMI (kg/m^2^)	1.385 (1.293–1.483)	<0.001	1.389 (1.296–1.488)	<0.001	-	-
BMI ≥ 30 kg/m^2^	16.009 (8.424–30.426)	<0.001	16.093 (8.454–30.633)	<0.001		
Age	0.992 (0.960–1.024)	0.617	-	-	-	-
DM	2.261 (0.921–5.555)	0.075	2.614 (1.013–6.742)	0.047	1.628 (0.615–4.310)	0.327
Dyslipidemia	1.675 (1.021–2.748)	0.041	1.819 (1.089–3.039)	0.022	1.006 (0.590–1.714)	0.983
HTN	0.996 (0.522–1.899)	0.989	-	-	-	-

Multivariate model: BMI ≥ 30 kg/m^2^ and age were adjusted as covariates. Abbreviations: BMI, body mass index; DM, diabetes mellitus; HTN, hypertension; MPVD, main portal vein diameter; OR, odds ratio; VPI, portal venous pulsatility index.

## Data Availability

The original contributions presented in this study are included in the article/Appendix A; further inquiries can be directed to the corresponding author.

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
