# Peer review of "The Portal Venous Pulsatility Index and Main Portal Vein Diameter as Surrogate Markers for Liver Fibrosis in Nonalcoholic Fatty Liver Disease and Metabolic-Dysfunction-Associated Steatotic Liver Disease"

_diagnostics, 2024, doi:10.3390/diagnostics14040393_

Round 1
Reviewer 1 Report
Comments and Suggestions for Authors
In the manuscript, entitled »Portal Venous Pulsatility Index as a Surrogate Marker for Liver Fibrosis in NAFLD and MASLD«, submitted to Diagnostics for a potential publication, the authors present their research article investigating the role of portal venous pulsatility index (VPI) in liver diagnostics. I am of opinion, that in the present form, the manuscript is not good enough to be published in Diagnostics. My comments to potentially improve it are enclosed.
The comments:
1. The study is retrospective. With how many examiners were ultrasound parameters measured? What wase intraobserver variability and how was determined? These results should be presented. In addition, both intra- and interobserver variabilities have to be presented for liver stiffness measurement.
2. What novelties the study brings? Namely, the study is retrospective, but the term MASLD is quite new. In addition, there were numerous studies, investigating VPI, performed.
3. The article is quite long with a lot of different data, making it not clear enough, which should be improved.
4. How many included patients were obese (3.1 basic caracteristics of included patients)?
5. How many patients had more than one cardiovascular risk factor? The presentation of results according to the latter would be of value.
6. Which statistical methods were used to asses the normality of investigated population? Which parameters were normally distributed?
7. How good the presented two multivariate models were? F statistics should also be presented.
8. References (and definitions) for liver fibrosis scores used in the article should be cited.
9. The proposed clinical algorithm for noninvasive diagnostics (laboratory and ultrasonographic) of liver disease in NAFLD and MASLD, according to the study results, would be of value, or at least the place of VPI in the diagnostic procedure.
Comments on the Quality of English Language
Minor editing of English language required.
Author Response
Response to reviewers
First of all, the authors express their sincere gratitude to the reviewers for their valuable and constructive critiques aimed at enhancing the quality of the manuscript. We have diligently addressed all the raised concerns and have made the necessary revisions, which are highlighted in red font within the revised manuscript. Please find below a comprehensive, point-by-point response to both the reviewers' and editor's comments. We hope that these revisions adequately meet their expectations.
Reviewer 1
In the manuscript, entitled »Portal Venous Pulsatility Index as a Surrogate Marker for Liver Fibrosis in NAFLD and MASLD«, submitted to Diagnostics for a potential publication, the authors present their research article investigating the role of portal venous pulsatility index (VPI) in liver diagnostics. I am of opinion, that in the present form, the manuscript is not good enough to be published in Diagnostics. My comments to potentially improve it are enclosed.
The comments:
- The study is retrospective. With how many examiners were ultrasound parameters measured? What was intraobserver variability and how was determined? These results should be presented. In addition, both intra- and interobserver variabilities have to be presented for liver stiffness measurement.
- We appreciate the insightful commentary provided by the reviewer. The ultrasound parameters were primarily measured by a single sonographer. Unfortunately, due to the retrospective nature of the study, the possibility of different operators conducting ultrasonography for the same patient in separate sessions was precluded. In response to this valid concern, we adopted a meticulous approach by involving two different sonographers to measure ultrasound parameters using sonographic imaging uploaded to the Picture Archiving and Communication System (PACS) (Line 353-360). The interobserver variability resulting from these measurements was calculated as 0.932 (95% CI: 0.900-0.952) for the Portal Venous Pulsatility Index (VPI) and 0.798 (95% CI: 0.376-0.911) for the Main Portal Vein Diameter (MPVD). While these figures indicate a promising level of reproducibility for these parameters, it is crucial to acknowledge certain limitations in our approach. Specifically, measurements by other sonographers were performed using only static photos, contributing to constraints in determining the calculated interobserver variabilities. Consequently, we have omitted statements emphasizing the reproducibility of these parameters as a strength in our study (line 429-432 in the previous version of the manuscript) and have instead incorporated a clarifying comment addressing this limitation in the revised manuscript's limitation section (line 449-452).
“Furthermore, despite the promising interobserver variabilities of ultrasound parameters in our study, measurements by other sonographers were not conducted in separate sessions and were solely based on static photos, introducing certain limitations regarding the calculated interobserver variabilities.”
- Regarding liver stiffness measurements, at least 10 valid measurements per patients was performed and median/interquartile range is documented in PACS. Therefore, we have stated the median value of median/interquartile range for liver stiffness measurement in the revised Table 3.1.
- In terms of intraobserver variability, we identified 107 and 166 ultrasound records with multiple measurements for VPI and MPVD, respectively. Utilizing these records, we assessed the intraobserver variability, which was found to be 0.959 (95% CI: 0.920-0.976) for VPI and 0.820 (95% CI: 0.450-0.920) for MPVD. These findings have been incorporated into the method and results section, lines 113-115 and lines 360-364, as articulated below.
“In cases where ultrasound parameters were measured multiple times, the median value was adopted for analysis.”
“In addition, intraobserver variability was assessed. A total of 107 and 166 sonographic records, each measured more than once for VPI and MPVD, respectively, were identified. Utilizing these records, intraobserver variability was determined to be 0.959 (95% CI: 0.920-0.976) for VPI and 0.820 (95% CI: 0.450-0.920) for MPVD.”
- What novelties the study brings? Namely, the study is retrospective, but the term MASLD is quite new. In addition, there were numerous studies, investigating VPI, performed.
- We thank the reviewer for giving us opportunity to elucidate the novel aspects of our study. As pointed out by the reviewer, the introduction of the term 'MASLD' signifies a recent nomenclature shift compared to NAFLD. Notably, while these terms may seem synonymous, MASLD necessitates at least one of five cardiometabolic risk factors, setting it apart from NAFLD. This distinction adds a novel aspect to our study, as it is the first to report the correlation between VPI and liver fibrosis in the context of MASLD.
In addition, our study, with a substantial sample size, adds valuable evidence to the ongoing discourse regarding the correlation between the VPI and liver fibrosis. The controversies observed in previous studies, demonstrated by the absence of correlation reported by Lu et al., might be attributable to relatively small sample sizes, potentially leading to type 2 errors. In contrast, our study, encompassing nearly 1,000 participants, establishes a robust foundation to elucidate the relationship between VPI and liver fibrosis.
Furthermore, our study introduces a unique approach by utilizing two ultrasound parameters, namely MPVD and VPI. The integration of these parameters in predicting significant fibrosis unveils the potential of these combined biomarkers as superior surrogates compared to individual assessments. We have incorporated a statement on this matter in the discussion section, specifically in lines 434-436:
“Lastly, we integrated MVPD with VPI in predicting significant fibrosis, showcasing the potential of these aggregated biomarkers as superior surrogate markers compared to each individually.”
- The article is quite long with a lot of different data, making it not clear enough, which should be improved.
- We appreciate the reviewer's observation regarding the extensive data presented in the article, which may hinder clarity. In response to this valid concern, we have revised Table 3.1 by removing parameters such as WBC, NLR, and CRP. These markers, primarily recognized as indicators of inflammation rather than fibrosis, were deemed non-essential for the focus of our study.
- How many included patients were obese (3.1 basic caracteristics of included patients)?
- In our study, 67.9% of the total population met the criteria for obesity, defined as a BMI of 25 or higher based on the consensus for the Asian population (Pan et al. Asia Pac J Clin Nutr. 2008;17(3):370-374/ World Health Organization; Regional Office for the Western Pacific. The Asia-Pacific perspective: redefining obesity and its treatment. Sydney: Health Communications Australia; 2000/ Kim et al. J Obes Metab Syndr. 2021;30(2):81-92). When stratified by significant fibrosis status, 93.5% of the SF (+) group had obesity, while 66.0% of the SF (-) group exhibited obesity. This information has been added to Table 3.1, and the results are presented in line 177 as follows:
“The mean BMI of the study population was 27.4 kg/m2 and the prevalence of metabolic comorbidities, such as obesity (67.9%), DM (4.4%), HTN (17.7%), and dyslipidemia (35.5%), was also assessed.”
- How many patients had more than one cardiovascular risk factor? The presentation of results according to the latter would be of value.
- In accordance with the reviewer's valuable suggestion, we have incorporated the number of patients with more than one cardiometabolic risk factor, which amounted to 568 individuals. Specifically, 275 patients exhibited three or more cardiometabolic risk factors. These results are now presented in Table 3.1 and detailed in the results section (3.1, line 179-182) as follows:
“Regarding the number of CMRFs present in each patient, 37.3% had one CMRF, 29.5% had two CMRFs, and 27.7% had three or more CMRFs.”
- Which statistical methods were used to assess the normality of investigated population? Which parameters were normally distributed?
- We appreciate the reviewer's insightful inquiry regarding the assessment of normality in our investigated population. To evaluate normality, we employed the Shapiro-Wilk test, as indicated in the revised methods section (Line 159-160):
“The normality of each variable was assessed using the Shapiro-Wilk test.”
- After a thorough review of all parameters utilized in this analysis, we found that all the laboratory variables exhibited a non-normal distribution, with a p-value less than 0.05 according to the Shapiro-Wilk test. In light of this identified non-normal distribution, necessary adjustments were made. Specifically, we employed the Wilcoxon rank-sum test for comparisons involving non-normally distributed parameters, as detailed in the revised method section (line 156). Consequently, to accurately represent variables not conforming to a normal distribution, Table 3.1 has been revised.
We acknowledge and apologize for any confusion caused by our initial presentation and express our sincere gratitude to the reviewer for bringing this critical flaw to our attention. In the revised baseline characteristics section (3.1), median values are now presented for non-normally distributed variables, as shown by the following lines:
Line 173: “Median age was 21.0 and the majority were male (98.0%).”
Line 205-208: “Median CAP score and median LSM for the entire cohort were 278.0 dB/m and 4.8 kPa, respectively, with both values being significantly higher in the SF (+) group compared to the SF (-) group (median CAP 340.0 vs. 271.0 dB/m, p < 0.001; median LSM 9.1 vs. 4.7, p < 0.001, SF (+) and SF (-) group, respectively).”
Line xxx-xxx: “In summary, the median age was 40.0, and 178 patients (96.7%) were male.”
- How good the presented two multivariate models were? F statistics should also be presented.
- We appreciate the reviewer for pointing out this issue. The Model 1 and Model 2 presented in Table 3 do not represent the multivariate analysis of all the variables listed in the table. Instead, these models were developed to reveal the correlation of specific variables with significant fibrosis, irrespective of the covariates included, namely Age and BMI. It is essential to clarify that these models are not intended to formulate a predictive model for significant fibrosis. Rather, they demonstrate the association of selected variables with significant fibrosis, accounting for age and BMI.
To enhance clarity and avoid potential confusion associated with the term 'model,' we have made revisions to the table. Instead of using the names Model 1 and Model 2, we now present Age-adjusted OR and Multivariable-adjusted OR. Additionally, we have removed the term 'univariate analysis' to further clarify the presentation.
- References (and definitions) for liver fibrosis scores used in the article should be cited.
è Regarding the cut-off values used in this study to determine significant fibrosis, references are cited in the manuscript with citation numbers 25 (Eddowes, P.J et al. Gastroenterology 2019, 156, 1717-1730, doi:10.1053/j.gastro.2019.01.042), 26 (Wong, V.W et al. Hepatology 2010, 51, 454-462, doi:10.1002/hep.23312), and 27 (Newsome, P.N. et al. Lancet Gastroenterol Hepatol 2020, 5, 362-373, doi:10.1016/s2468-1253(19)30383-8.). For non-invasive indices for liver fibrosis and steatosis, we have added the following references of these scores with the citation numbers ranging from 29 to 32 (line 147):
[29] FIB-4 score: Sterling RK et al. Hepatology. 2006;43(6):1317-1325. doi:10.1002/hep.21178
[30] NAFLD Fibrosis score: Angulo P et al. Hepatology. 2007;45(4):846-854. doi:10.1002/hep.21496
[31] APRI score: Wai CT et al. Hepatology. 2003;38(2):518-526. doi:10.1053/jhep.2003.50346
[32] Hepatic Steatosis Index: Lee JH et al. Dig Liver Dis. 2010;42(7):503-508. doi:10.1016/j.dld.2009.08.002
We thank the reviewer again for pointing this out, which has enhanced the clarity of the paper.
- The proposed clinical algorithm for noninvasive diagnostics (laboratory and ultrasonographic) of liver disease in NAFLD and MASLD, according to the study results, would be of value, or at least the place of VPI in the diagnostic procedure.
We appreciate the reviewer's insightful comments on our proposed biomarkers for non-invasive diagnostics in liver disease associated with NAFLD and MASLD. As discussed in our manuscript, VPI is an easily measurable parameter that can be performed at the point of care, potentially playing a role in clinical practice to discriminate NAFLD patients at higher risks. Despite the limitations inherent in our study, we believe that the results derived from our research could contribute to the consideration of VPI in the algorithm for diagnosing significant fibrosis.
Reviewer 2 Report
Comments and Suggestions for Authors
This article titled as " Portal venous pulsatility index as surrogate marker for liver fibrosis in NAFLD and MASLD" is aimed to identify easily measurable ultrasound parameters capable of predicting liver fibrosis in patients with NAFLD and MASLD. This article is well written. But I have a few suggestions to make it a better piece.
1, In this paper, liver fibrosis was diagnosis by liver stiffness measurement. As mentioned by the authors, although LSM has been proved to correlated with liver fibrosis, histological data is the golden diagnosis for significant fibrosis. And significant fibrosis is defined as F2 or greater in METAVIR. METAVIR is a classification system based on histological data. Using LSM data here might cause confusion and misdiagnosis. Measurement of LSM and VPI at the same time could induce bias more easily. In Lu`s study (PMID:35251901), no correlation was detected between VPI and NAFLD in patients with biopsy-proved NAFLD. So, I suggest to add histological data in this study.
2、In this article, VPI and MPVD are both detected as biomarkers correlated with liver fibrosis. And the conclusion of this article is that VPI and MPVD are useful in predicting significant fibrosis in patients with NAFLD and MASLD. However, the title of this article only mentioned VPI. I suggest to make a little change here.
3、In this article, a nomogram was developed to predict significant liver fibrosis in NAFLD. However, only 68 patients with significant fibrosis was included in the cohort. And these patients had to been divided into training and test cohorts. I doubt if the number of patients with significant fibrosis is too small.
4、The aim of this article is to find a useful noninvasive biomarker to predict significant fibrosis in patients with NAFLD and MASLD. LSM is a non invasive biomarker for detecting fibrosis. And in this paper, LSM is used as a golden standard for liver fibrosis. So the meaning to find another useful noninvasive biomarker might be limited. I suggest the diagnostic value for liver fibrosis of VPI , MPVD or nomogram should be compared with LSM.
Comments on the Quality of English LanguageThe quality of English is good. And only minor editing is needed.
Author Response
Response to reviewers
First of all, the authors express their sincere gratitude to the reviewers for their valuable and constructive critiques aimed at enhancing the quality of the manuscript. We have diligently addressed all the raised concerns and have made the necessary revisions, which are highlighted in red font within the revised manuscript. Please find below a comprehensive, point-by-point response to both the reviewers' and editor's comments. We hope that these revisions adequately meet their expectations.
Reviewer 2
This article titled as " Portal venous pulsatility index as surrogate marker for liver fibrosis in NAFLD and MASLD" is aimed to identify easily measurable ultrasound parameters capable of predicting liver fibrosis in patients with NAFLD and MASLD. This article is well written. But I have a few suggestions to make it a better piece.
- In this paper, liver fibrosis was diagnosis by liver stiffness measurement. As mentioned by the authors, although LSM has been proved to correlated with liver fibrosis, histological data is the golden diagnosis for significant fibrosis. And significant fibrosis is defined as F2 or greater in METAVIR. METAVIR is a classification system based on histological data. Using LSM data here might cause confusion and misdiagnosis. Measurement of LSM and VPI at the same time could induce bias more easily. In Lu`s study (PMID:35251901), no correlation was detected between VPI and NAFLD in patients with biopsy-proved NAFLD. So, I suggest to add histological data in this study.
- We sincerely appreciate the constructive comment provided by the reviewer. Unfortunately, our study had a limited number of patients undergoing liver biopsy, with only two individuals having documented histological data in this cohort. One patient exhibited F1 in the METAVIR classification system, with LSM and VPI values of 7.2 kPa and 0.26, respectively. Another patient also showed F1 in the METAVIR classification system, with LSM and VPI values of 8.0 kPa and 0.24. Given the distinctive demographic attributes of our study population, primarily composed of individuals in their early twenties who are obligated to serve in the military due to the mandatory conscription system in South Korea, persuading them to undergo liver biopsy presented notable challenges. The difficulties stem from the fact that these young Korean men find themselves in a vulnerable position, compelled to fulfill their military duty, which may render them more susceptible to external pressures and less inclined to willingly undergo medical procedures such as liver biopsy.
While we acknowledge the contrasting results presented by Lu et al., it's noteworthy that Baikpour et al. (AJR Am J Roentgenol. 2020;214(4):786-791. doi:10.2214/AJR.19.21963), utilizing biopsy-proven NAFLD, reported a significant correlation between significant fibrosis and VPI. Additionally, Hamed (The Egyptian Journal of Hospital Medicine. 2022;86(1): 506-512), using shear wave elastography similar to our tool (FibroScan), demonstrated a significant correlation between significant fibrosis and VPI. Despite the limitations of our study, such as the scarcity of biopsy data, we believe our research offers novelty with a relatively larger cohort compared to studies conducted with biopsy-proven NAFLD.
- In this article, VPI and MPVD are both detected as biomarkers correlated with liver fibrosis. And the conclusion of this article is that VPI and MPVD are useful in predicting significant fibrosis in patients with NAFLD and MASLD. However, the title of this article only mentioned VPI. I suggest to make a little change here.
- We sincerely appreciate the valuable comment provided by the reviewer, which enhances the clarity of the implications in our study. We agree with the reviewer’s opinion that MPVD should also be included in the title. Consequently, we have revised the title as follows:
“Portal Venous Pulsatility Index and Main Portal Vein Diameter as Surrogate Markers for Liver Fibrosis in NAFLD and MASLD”
- In this article, a nomogram was developed to predict significant liver fibrosis in NAFLD. However, only 68 patients with significant fibrosis was included in the cohort. And these patients had to been divided into training and test cohorts. I doubt if the number of patients with significant fibrosis is too small.
- We appreciate the reviewer for providing us the opportunity to address this concern. Indeed, our study included only 68 patients with significant fibrosis, with 43 and 25 patients allocated to the training and test cohorts, respectively. We acknowledge that this number of events might be perceived as insufficient for building a model by some readers. However, it's important to note that the 'Rule of Ten Events per Variable' has been widely adopted for regression analysis, as suggested in a previous statistical journal (Peduzzi P et al. J Clin Epidemiol. 1996;49(12):1373-1379. doi:10.1016/s0895-4356(96)00236-3). Additionally, other studies on statistics have proposed relaxing this rule to allow 5-9 events per variable (Vittinghoff E et al. Am J Epidemiol. 2007;165(6):710-718. doi:10.1093/aje/kwk052).
In our study, we utilized two variables, namely VPI and MPVD, to predict significant fibrosis, and we ensured that the number of events exceeded the rule of ten in both the training and test cohorts. While external validation is necessary to further establish the predictive value of the proposed biomarkers, we believe that the number of events in our study is sufficient to derive meaningful insights from these biomarkers.
- The aim of this article is to find a useful noninvasive biomarker to predict significant fibrosis in patients with NAFLD and MASLD. LSM is a non invasive biomarker for detecting fibrosis. And in this paper, LSM is used as a golden standard for liver fibrosis. So the meaning to find another useful noninvasive biomarker might be limited. I suggest the diagnostic value for liver fibrosis of VPI , MPVD or nomogram should be compared with LSM.
- In order to compare the diagnostic performance of VPI and MPVD with LSM, it is crucial to change the standard method for assessing liver fibrosis from LSM to liver biopsy in our study. Regrettably, our study cohort lacked sufficient histologic data, rendering it unfeasible to conduct this specific analysis. However, we fully acknowledge the merit of the reviewer's suggestion in enhancing the depth of the study results. To acknowledge this limitation and emphasize the potential for future research, we have added a sentence to the limitation section, line 443-445, as follows:
“A comparison of predictive performance between LSM and VPI in biopsy-proven cohorts could enhance the significance of VPI in predicting significant fibrosis.”
Once again, we extend our gratitude to the reviewers and the editor for their valuable feedback, which has significantly improved the manuscript. We hope that the revised version now meets the high standards expected for publication.
Round 2
Reviewer 1 Report
Comments and Suggestions for Authors
I have carefully read the revised manuscript entitled »Portal Venous Pulsatility Index as a Surrogate Marker for Liver Fibrosis in NAFLD and MASLD« submitted to Diagnostics for a potential publication. The manuscript has been substantially improved, taken practically all my suggestions and comments into consideration. Anyway, I also proposed to include a figure, presenting the algorithm of diagnostics in MASLD, according to their study results.
Comments on the Quality of English Language
Minor editing of English language required
Author Response
- We express our gratitude to the reviewer for their valuable comment, which significantly contributed to the enhancement of our study. In response to this suggestion, we have incorporated Figure 5 into the manuscript, illustrating the proposed algorithm for diagnosing liver fibrosis in MASLD patients using ultrasound parameters. Furthermore, we have included a corresponding paragraph in the manuscript (lines 352-354) to highlight the addition of this figure:
“Using the provided ultrasound parameters, an algorithm for assessing liver fibrosis can be proposed, with respective thresholds of 120 for the nomogram (sensitivity 71.9%, specificity 74.7%) and 0.26 for the VPI (sensitivity 59.5%, specificity 77.8%) (Fig. 5).”
Reviewer 2 Report
Comments and Suggestions for Authors
We appreciated the response from the authors. The revision has met our exception. No further revision is needed.
Author Response
- We sincerely appreciate the reviewer's feedback and are delighted to hear that the revised manuscript has met their expectations. Thank you for taking the time to review our work, and we are grateful for your positive assessment.